# Research on video adversarial attack with long living cycle

**Zeyu Zhao**[1]  **Ke Xu**[1]  **Xinghao Jiang**[1]  **Tanfeng Sun**[1]

[1] Shanghai Jiao Tong University, China
**329161318zzy, 113025816, xhjiang, tfsun@sjtu.edu.cn**

## Abstract

In recent years, the vulnerability of networks has attracted the attention of researchers. However, in these methods, the impact of video compression coding on the added adversarial perturbation, i.e., the robustness of the video adversarial example, is not considered. When an adversarial sample is just generated, its attack capability is the strongest. However, with multiple video encoding and video decoding in Internet transmission, the added adversarial disturbance will be continuously eliminated, eventually leading to the attack on the adversarial sample performance disappearing. We define this phenomenon as the decay of the lifetime of adversarial examples. We propose an adversarial attack method based on optimized integer space to resist this performance degradation. The robustness of anti-coding, the visual concealment, and the attack success rate are all considered during the attack process. In addition, we have also reduced the rounding loss caused by normalization in the deep neural network model process. The contributions of our methods are 1) We show the performance degradation caused by video compression coding on existing video adversarial attack methods, which seems an effective way for detecting of defending video adversarial examples. 2) A robust video adversarial attack method is proposed to resist video compression coding. The experiment shows that our method performs better on the robustness of anti-coding, visual concealment, and attack success rate.

## 1 INTRODUCTION

In recent years, Deep Neural Networks (DNNs) have achieved remarkable results in many computer-vision fields. It has amounts of research, and very good performance in the fields of image classification Mikoajczyk and Grochowski [2018], and video recognition Tran et al. [2018], Hara et al. [2018]. However, the appearance of adversarial examples makes people begin to think of the vulnerability and robustness of DNNs Hussain et al. [2021], Goodfellow et al. [2015]. Adding some subtle disturbances to the clean samples can make the original well-performing DNN models misclassify or even fail to correctly identify the samples. This phenomenon can be found in many fields, such as image classification Goodfellow et al. [2015], video recognition Wei et al. [2019], Chen et al. [2021], and human skeleton recognition Wang et al. [2021].

In fact, during the transmission and storage of videos, video compression coding is necessary. When an adversarial sample is just generated, its attack capability is the strongest, but with multiple video encoding and video decoding in the process of Internet transmission, the added adversarial disturbance will be continuously eliminated, eventually leading to the attack of the adversarial sample performance disappears. We define this phenomenon as the decay of the lifetime of adversarial examples. For normal samples, the loss generated in the video compression coding and decoding process can be considered Gaussian noise, such a difference is within an acceptable range, and a well-trained DNN model can resist this white noise. However, for adversarial examples, in the process of adversarial perturbation generating, the scale of perturbation is designed to be very small, which causes the adversarial disturbance to be eliminated very easily.

In our experiment, we found that almost all previous works have shown vulnerability in the face of video compression coding. The attack success rate of coded samples has decreased by more than half compared to the original adversarial examples. To resist this performance degradation, we propose an adversarial attack method based on optimized integer space. The robustness of anti-coding, the visual concealment, and the attack success rate are all considered during the process of attack.

*Accepted for the 38th Conference on Uncertainty in Artificial Intelligence* (UAI 2022).

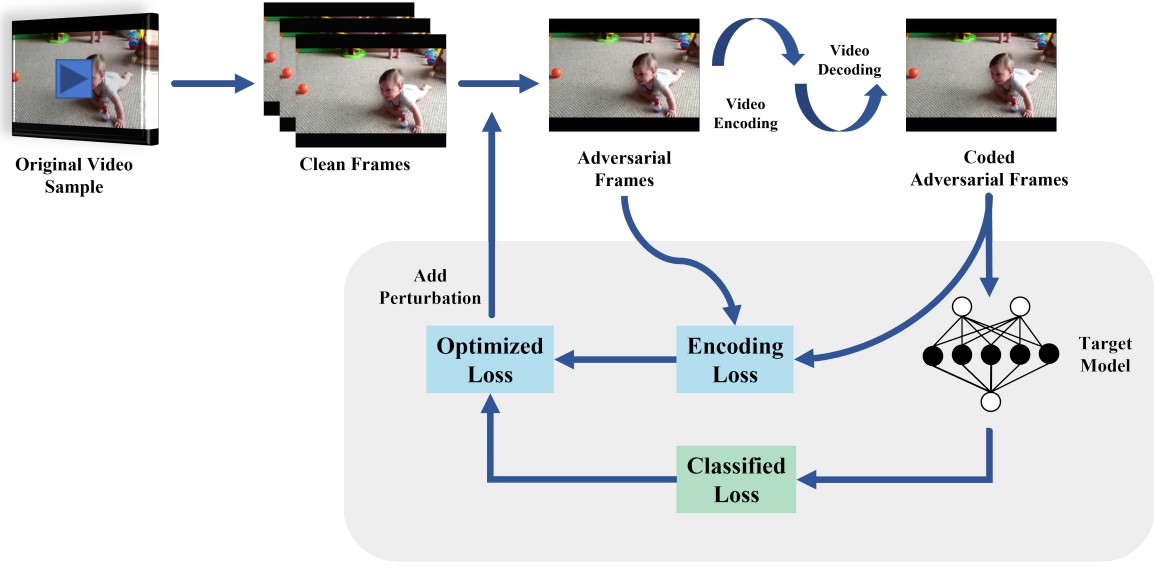

**Figure 1: Anti-Encoding Adversarial Attack**

Making full use of these observations, we present a novel optimization-based attack method named Anti-encoding Adversarial Attack and demonstrate its effectiveness on the video recognition task and high robust capability in the face of video compression coding.

In detail, we took a very simple but reasonable approach: as shown in Figure1, when we iteratively update the adversarial examples through the gradient information of the optimization function, we will perform a video encoding and decoding work on the adversarial examples every epoch and input the decoded video frames as samples into the model for testing. If the attack can be successful, then we will retain this video sample as the final adversarial example. If it is unsuccessful, we will use the gradient information of the sample to update the next round until the attack is successful or the maximum number of iterations is reached. In this way, we can ensure that the adversarial examples generated each time are resistant to video compression encoding.

We evaluate our approach on the popular video recognition dataset: UCF-101 Soomro et al. [2012], we select two victim models form a wide range of video recognition methods: I3D on ResNet Carreira and Zisserman [2017] and R(2+1)D Contributors [2020], each model can achieve high accuracy in UCF-101 dataset. We can get all the gradient information from the model, which means our approach is a type of white-box attack. The main contribution of this work is as follows:

- We show the performance degradation caused by video compression encoding on existing video adversarial attack methods, which seems an effective way of detecting and defending video adversarial examples.

- A robust video adversarial attack method is proposed to resist video compression coding. The experiment shows that our method achieves better performance on the robustness of anti-coding, visual concealment, and attack success rate.

## 2 RELATED WORK

The related work comes from two aspects: video action recognition and adversarial attack.

### 2.1 VIDEO RECOGNITION WITH DEEP LEARNING

The great performance of the Deep Neural Network on static image tasks motivates researchers to use it on the video task. Compared to static images, the video can be seen as a list of images. The images only have 2-dimension information, and videos have 3-dimension information; the timing feature is the key to the video action recognition mission. Inspired by the image classification task, many researchers replaced 2D convolutional kernels with 3D convolutional kernels but did not change the network structure, and they got great performance on the video recognition task Hara et al. [2018], Carreira and Zisserman [2017]. But the number of parameters needed to be trained is too large, so another method is just using a 3D convolutional kernel at the low layer of the whole network and using the 2D convolutional kernel to extract features at the higher layer Tran et al. [2018]. Both methods can have a good performance

on the video recognition task. In this paper, we choose R3D and R(2+1)D as our target models, and we use the open-source mmaction2 Contributors [2020]video recognition platform as our experiment platform. Our methods can be verified easily on this platform.

## 2.2 ADVERSARIAL ATTACK

While deep neural networks achieve excellent performance in multiple areas of computer vision, the appearance of adversarial examples makes people think more about DNN's robustness and safety. By adding some small disturbances that are imperceptible to the human eyes in a clean sample, the target model cannot classify the samples correctly. There are several methods for generating adversarial image examples. Fast gradient sign method (FGSM) Goodfellow et al. [2015] uses the gradient information to update the input examples and attack the neural network successfully. Projected gradient descent (PGD) Madry et al. [2018] algorithm also uses gradient information to update clean samples until the samples can be classified wrong. Unlike FGSM, PGD considers the nonlinear feature of CNN, and it uses more but smaller update steps to generate adversarial examples. Besides, the C&W algorithm Carlini and Wagner [2017]is a kind of optimization-based adversarial attack method. This algorithm regards classification loss and disturbance amplitude as optimization targets and reduces the range of disturbance as much as possible while attacking the model. The adversarial examples generated by this method have good concealment in vision.

The generation of video adversarial examples is very similar to adversarial image examples. In Sparse Adversarial Perturbations for Videos Wei et al. [2019], the researchers attacked the CNN+RNN model and proposed a kind of optimization-based adversarial attack. In Appending Adversarial Frames for Universal Video Attack Chen et al. [2021], researchers appended dummy frames at the end of the video samples, and they only added perturbation on the dummy frames. They claimed this method could generate video adversarial examples with very high transferability. Both methods can generate adversarial examples with a high attack success rate, but none of them consider the impact of video compression encoding. They only test their videos in the floating number domain. That means examples generated by these methods cannot transfer stability.

## 3 METHODOLOGY

In this section, we will introduce our anti-coding adversarial attack algorithm. Our method is an optimization-based approach.

## 3.1 OPTIMIZATION PROBLEM DESCRIPTION

Let $X \in [0, 255]^{T \times C \times W \times H}$ denotes a clean video sample, where the $T$ denotes the number of frames, $C$ denotes color channels, $W$ and $H$ denote width and height of the video. And $\hat{X}$ is the adversarial video example in the floating number domain, and $X_c$ represents the encoded video adversarial example. Let $E = \hat{X} - X$ denotes the added perturbations in the adversarial, and $E_c = \hat{X}_c - \hat{X}$ denotes the video compression encoding loss. So, our optimization question can be presented as Eq.1:

$$
\begin{aligned}
argmin_E ||E||_{2,1} + ||E_c||_{2,1} \\
- l(y_{truth}, y_{select}, f(\hat{X}))
\end{aligned}
\tag{1}
$$

Where $l(\cdot)$ denotes the classification loss function, in this paper, we choose the loss function as below:

$$
\begin{aligned}
l(y_{truth}, y_{select}, f(\hat{X})) = \\
max(0, f(\hat{X})_{y_{truth}} - f(\hat{X})_{y_{select}})
\end{aligned}
\tag{2}
$$

The $y_{truth}$ is the truth label of a sample $X$, $y_{select}$ is an optional label, and we found that if we choose a target label as the optimization target label, the optimization function can converse more quickly. But this is not a targeted attack because we just use the selected label to accelerate the convergence of the optimization function. The choice of labels can be random. We will introduce our target attack later. In our experiment, we choose the label with the second-highest confidence score. The $f(\cdot)$ is the victim video recognition model.

## 3.2 ROUNDING LOSS

For most video recognition DNN models, during their training process, they normalize the video samples from the integer domain to the float number domain. In this way, the model can have a fast convergence. So, previous adversarial examples generating methods also generate adversarial examples in floating number space. In our experiment, we found that if we want to save our adversarial images, the rounding loss caused by normalization cannot be ignored. Assume the original sample $X \in [0, 255]$ and the sample after normalization $X_{norm} = \frac{X}{255} \in [0, 1]$we add the adversarial perturbation in $X_{norm}$ : $\hat{X}_{norm} = X_{norm} + P_{norm} \in [0, 1]$, then if we storage our adversarial example by video encoding, the adversarial example is $\hat{X} = Round(\hat{X}_{norm} \times 255) \neq \hat{X} \times 255$. So, if we want to test $\hat{X}$ again, the result may be different, like$f((\frac{\hat{X}}{255})) \neq f(\hat{X}_{norm})$. To avoid this, in this paper, we add our perturbation in the original integer space, $\hat{X} = X + P \in [0, 255]$. In this way, we can easily control the scale of perturbation and retain all the perturbation.

## 3.3 VIDEO COMPRESSION ENCODING

The process of video compression encoding can be presented as below:

$$X_c = DCT(X)Q \qquad (3)$$

The function $DCT(\cdot)$ denotes discrete cosine transform and converts the video frames from the spatial domain to the frequency domain. This step will not cause loss. The $Q$ denotes the quantization matrix of the compression step. The lossless steps in the process of encoding will not be shown in our algorithm. Next, we will use a simple example to explain why the quantization step will bring loss.

This is easy to understand, This is easy to understand. If there are four different pixels, 33, 34, 35, 36. After quantized division, it will become 1,1,1,1. After inverse quantization, these four values will become the same, all of which are 32. This causes a great loss of information, shown in the image is the reduction of color space and the lack of details.

When the encoded video needs to be extracted frames, the process can be presented as eq.4:

$$X' = IDCT(X_c Q^{-1}) \qquad (4)$$

Where the $IDCT$ denotes inverse DCT, in this process, the output data $X'$ is different with $X$, there exists a difference $E_c = X' - X \neq 0$.

---

**Algorithm 1** Anti-encoding Adversarial Attack

**Input:** original video sample $X$, target model $f(\cdot)$, truth label $t_{truth}$
**Output:** video adversarial example $\hat{X}$

1: scale $\leftarrow 1$
2: $\hat{X} = Clip(X+P,0,255)$ where $P \sim U(-scale,scale)$
3: $y_{select}$=Random(0,N) and $y_{select} \neq y_{true}$, where N is the number of classes
4: **while** attack not success: **do**
5:  $\hat{X}_c$=VideoDecode(VideoEncode($\hat{X}$))
6:  $E_c = \hat{X}_c - \hat{X}$
7:  $\hat{X}_{norm} = Normalize(\hat{X}_c)$
8:  $Z = f(\hat{X}_{norm})$
9:  Calculate loss function in eq.5
10:  $g = \frac{\partial Loss}{\partial \hat{X}}$
11:  P = P-sign(g)
12:  $\hat{X}$= Clip(X+P,0,255)
13: **end while**

---

## 3.4 ADVERSARIAL ATTACK

According to the above conditions, we can construct our untargeted attack optimization loss function:

$$Loss(X,\hat{X}) = \lambda_1 ||X - \hat{X}||_{2,1} + \lambda_2 ||\hat{X} - \hat{X}'||_{2,1}$$
$$- max(0, f(\hat{X}_{norm})_{y_{truth}} - f(\hat{X}_{norm})_{y_{select}}) \qquad (5)$$

In untargeted attack, We chose the label with the second highest classification confidence as the target for optimization. The purpose of this is to speed up the attack as much as possible.

For targeted attack, the optimization loss function can be presented as below:

$$Loss(X,\hat{X}) = \lambda_1 ||X - \hat{X}||_{2,1} + \lambda_2 ||\hat{X} - \hat{X}'||_{2,1}$$
$$- max(0, f(\hat{X}_{norm})_{y_{target}} - max(f(\hat{X}_{norm}))) \qquad (6)$$

Where $\lambda_1$ and $\lambda_2$ are balance parameters. $||E||_{2,1}$ denotes $L_{2,1} - Norm$, it can be a good measure of the degree of perturbation added to the video adversarial examples. The calculation method is as follows:

$$||E||_{2,1} = \sum_i^T |E_i|_2 \qquad (7)$$

Where $E_i$ denotes the perturbation added at the $i_{th}$ frame. And the $|E_i|_2$ represents the $l_2 - norm$ of $E_i$, then we can ensure that all the perturbations can be optimized through this way.

The whole process of the Anti-coding Adversarial Attack algorithm is shown in Algorithm 1.

## 4 EXPERIMENT AND ANALYSIS

In this section, we will analyze our experimental methods and experimental results and compare them with previous white-box attack adversarial example generation methods.

### 4.1 EXPERIMENT SETTING:

**Dataset:** we choose UCF-101 dataset Soomro et al. [2012]as our test dataset, the videos in UCF-101 are collected from website and contain 101 kinds of actions. It contains 13320 realistic action videos. We choose 9600 videos as the train set and 3720 videos as the test set to train our threat video recognition model.

**Video recognition model:** we selected two models as the target model: I3D on ResNet Tran et al. [2018] and R(2+1)D Hara et al. [2018]. The models we choose are all pretrained on the kinetic-400 dataset and then fine-tuned

by the UCF-101 dataset. In order to ensure that the experimental results are easy to reproduce, we adopted the open-source mmaction2 platform as the test platform. It is easy to reconstruct our algorithm. And achieved similar accuracy rates to the reputation of proposing these model papers.

**Attack settings:** for a single video attack, we set the number of input frames as 32, and we attack all the frames. For the FGSM attack, we set the maximum perturbation amplitude to 15/255. For the PGD attack and sparse video attack, we set the max iterations as 100, and the max amplitude is 5/255. We only choose videos that can be classified correctly as target samples, and over 500 videos have been selected during each experiment. For video compression encoding methods, we choose MPEG4, h264, and HEVC algorithms. The MPEG4 is the encoding algorithm of the UCF-101 dataset, and h264 is the widely used encoding method on websites, while the HEVC is the newest video compression encoding standard. All of those encoding methods have been widely used in industry and academia.

**Metrics:** refer to the sparse attack paper Wei et al. [2019], we use four metrics to evaluate various aspects.

**Fooling ration before encoding (F):** is defined as the percentage of non-encoded adversarial videos that are successfully misclassified.

**Perceptibility (P):** denotes the average scale of perturbations added into the adversarial examples. In this paper, we use $L_\infty - norm$ to measure visual concealment.

**L2 Norm:** represents the L2 norm of the added perturbation. Although we have used the maximum perturbation amplitude, when the maximum perturbation amplitude is the same, the L2 norm can well represent the degree of the added perturbation. When calculating the L2 norm, we use the normalization process; that is, no matter whether the original perturbation is added in the integer RGB space or the floating-point number space between [0,1], we will scale it to between 0 and 1 for the calculation.

**Attack success rate (ASR):** the final attack success rate of encoded videos, and this metric can represent the ultimate attack capability of the adversarial sample.

**Resistance (R):** It represents the ability to resist video coding damage. For example, there are M non-encoded adversarial examples that can attack the model successfully. After video encoding, only $M'(M' \le M)$ videos can still attack successfully, and the prediction label is the same as the prediction label before encoding, then $R = M'/M$.

## 4.2 ATTACK PERFORMANCE

### 4.2.1 Untargeted attack:

In Table 1, we show the attack effect of the untargeted attack adversarial examples before video compression encoding. In Table 2, we show the attack effects of different algorithms after video compression encoding and the robustness against video compression encoding.

Obviously, our method has a good performance in resisting video encoding. Other iteration-based methods: like PGD and sparse attack, although they can achieve a high attack success rate on non-encoded adversarial examples, cannot retain good performance in the face of encoded videos.

| Attack Methods | Target Model | Video Encoding Methods | F/% | P/% | L2 Norm |
|---|---|---|---|---|---|
| FGSM | R3D | MPEG4 | 97.80 | 5.88 | 150.50 |
| | | H264 | 97.80 | 5.88 | 150.50 |
| | | HEVC | 97.80 | 5.88 | 150.50 |
| | R(2+1)D | MPEG4 | 99.00 | 5.88 | 152.39 |
| | | H264 | 99.00 | 5.88 | 152.39 |
| | | HEVC | 99.00 | 5.88 | 152.39 |
| PGD | R3D | MPEG4 | 99.80 | 1.35 | 31.21 |
| | | H264 | 99.80 | 1.35 | 31.21 |
| | | HEVC | 99.80 | 1.35 | 31.21 |
| | R(2+1)D | MPEG4 | 100 | 1.25 | 31.64 |
| | | H264 | 100 | 1.25 | 31.64 |
| | | HEVC | 100 | 1.25 | 31.64 |
| Sparse Attack | R3D | MPEG4 | 98.40 | 1.20 | 30.41 |
| | | H264 | 98.40 | 1.20 | 30.41 |
| | | HEVC | 98.40 | 1.20 | 30.41 |
| | R(2+1)D | MPEG4 | 99.20 | 1.21 | 30.95 |
| | | H264 | 99.20 | 1.21 | 30.95 |
| | | HEVC | 99.20 | 1.21 | 30.95 |
| Ours | R3D | MPEG4 | 96.60 | **1.17** | **21.54** |
| | | H264 | 96.80 | **1.16** | **21.26** |
| | | HEVC | 98.50 | **1.18** | **21.55** |
| | R(2+1)D | MPEG4 | 99.20 | **1.18** | **20.56** |
| | | H264 | 99.40 | **1.16** | **21.55** |
| | | HEVC | 99.45 | **1.18** | **20.12** |

Table 1: Untargeted Attack

In untargeted attacks, the previous methods can still retain a high attack success rate after video compression encoding. However, this doesn't mean they have high resistance. In our experiment, we found that the predicted label after video encoding is different from the predicted label before video encoding. For example, the truth label is 1, and the

predicted label of adversarial example before video encoding is 2, but after video encoding, the predicted label of adversarial example maybe 3 or 4. Although we can use this coded sample to attack success, the robustness of the adversarial example has been destroyed. The process of video encoding will add unpredictable noise to the video. Therefore, the predicted label of the video sample will deviate to an unknown place. In the discussion of targeted attacks, this phenomenon will be further demonstrated and discussed.

| Attack Methods | Target Model | Video Encoding Methods | ASR/% | R/% |
|---|---|---|---|---|
| FGSM | R3D | MPEG4 | 96.20 | 61.76 |
| | | H264 | **97.00** | 67.48 |
| | | HEVC | 97.00 | 67.48 |
| | R(2+1)D | MPEG4 | 98.60 | 67.07 |
| | | H264 | 98.80 | 67.27 |
| | | HEVC | 98.60 | 66.67 |
| PGD | R3D | MPEG4 | 91.40 | 69.94 |
| | | H264 | 91.80 | 71.34 |
| | | HEVC | 88.00 | 65.53 |
| | R(2+1)D | MPEG4 | 96.20 | 58.00 |
| | | H264 | 96.00 | 58.00 |
| | | HEVC | 95.40 | 56.00 |
| Sparse Attack | R3D | MPEG4 | 89.80 | 70.33 |
| | | H264 | 89.60 | 70.12 |
| | | HEVC | 85.80 | 65.45 |
| | R(2+1)D | MPEG4 | 95.20 | 57.46 |
| | | H264 | 95.40 | 57.46 |
| | | HEVC | 94.60 | 55.44 |
| Ours | R3D | MPEG4 | **96.60** | **100** |
| | | H264 | 96.80 | **100** |
| | | HEVC | 98.50 | **100** |
| | R(2+1)D | MPEG4 | **99.20** | **100** |
| | | H264 | **99.40** | **100** |
| | | HEVC | **99.45** | **100** |

Table 2: Video encoding untargeted attack

Besides, our method can use smaller perturbations to get the highest success rate. We think this is the advantage of putting the perturbation in the integer space. When we place the disturbance in the integer space, the normalized data will be more like the normal sample and have the same distribution with clean samples. And if the disturbance is placed in the floating-point number space, the amplitude of each update is difficult for us to control. While the rounding loss is caused, the distribution differs from the original sample and is also difficult to control. In the control of the attack amplitude, it will be more difficult.

It should be noted that in the experiment, our method will change the perturbation amplitude under different video encoding methods, while other methods will not change. This is because, in our adversarial example generation method, we will take video encoding into consideration. Therefore, different video encoding methods will bring different attack performances. The previous methods did not take this into consideration, only the video coding test was added in the final link, so there will be no change.

### 4.2.2 Targeted Attack:

Next, we will analyze the experimental data of the targeted attack. The results of the target attack are shown in Table 3 and Table 4.In Table 3, we show the attack effect of the targeted attack adversarial sample before video compression encoding. In Table 4, we show the attack effects of different algorithms after video compression encoding and the robustness against video compression encoding.

| Attack Methods | Target Model | Video Encoding Methods | F/% | P/% | L2 Norm |
|---|---|---|---|---|---|
| Sparse Attack | R3D | MPEG4 | 98.00 | 1.96 | 31.34 |
| | | H264 | 98.00 | 1.96 | 31.34 |
| | | HEVC | 98.00 | 1.96 | 31.34 |
| | R(2+1)D | MPEG4 | 98.50 | 1.96 | 33.02 |
| | | H264 | 98.50 | 1.96 | 33.02 |
| | | HEVC | 98.50 | 1.96 | 33.02 |
| Ours | R3D | MPEG4 | **100** | **1.92** | 31.56 |
| | | H264 | **100** | **1.94** | 33.59 |
| | | HEVC | **100** | **1.89** | 32.78 |
| | R(2+1)D | MPEG4 | **100** | **1.96** | 33.16 |
| | | H264 | **100** | **1.92** | 34.11 |
| | | HEVC | **100** | **1.92** | 31.56 |

Table 3: Targeted Attack

For the setting of the targeted attack, we selected the sparse attack to compare with our experiment. This is because only the optimized-based adversarial examples generation methods can control the target of the attack. In the experiment, we selected label "1" as the target, and the selected samples were all classified correctly. The samples with the true label of 1 have a target of 2.

In Table 3, we can see that before encoding, the two methods have relatively good attack effects and have similar attack amplitudes. This is very reasonable because the difficulty of a targeted attack is significantly greater than that of an untargeted attack. In order to achieve a better attack effect, the magnitude of the attack disturbance will be slightly larger than that of an untargeted attack, but it is still kept at a relatively small level. Our method has a slightly higher

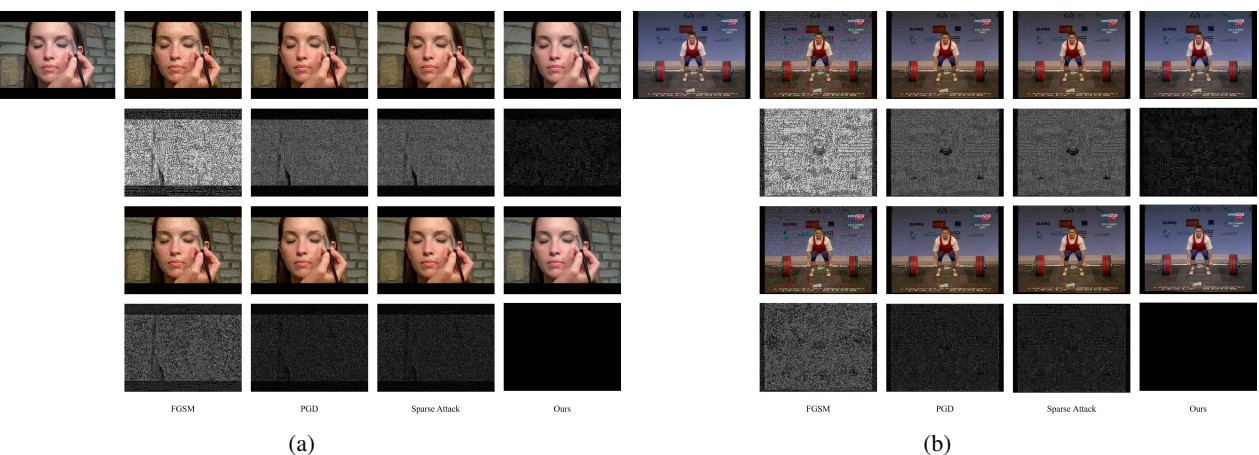

(a)                                                    (b)

Figure 2: Untargeted Attack, from left to right are clean image, FGSM, PGD, sparse attack and our method. The first line is the adversarial examples and clean sample, the second line is the added perturbation, the third line is the adversarial examples after video encoding, and the final line is the difference between non-coded image and coded image

| Attack Methods | Target Model | Video Encoding Methods | ASR/% | R/% |
|---|---|---|---|---|
| Sparse Attack | R3D | MPEG4 | 4.50 | 4.59 |
| | | H264 | 9.50 | 9.69 |
| | | HEVC | 7.50 | 7.65 |
| | R(2+1)D | MPEG4 | 23.50 | 23.86 |
| | | H264 | 29.00 | 29.44 |
| | | HEVC | 20.50 | 20.81 |
| Ours | R3D | MPEG4 | **96.60** | **100** |
| | | H264 | **100** | **100** |
| | | HEVC | **100** | **100** |
| | R(2+1)D | MPEG4 | **100** | **100** |
| | | H264 | **100** | **100** |
| | | HEVC | **100** | **100** |

Table 4: Video encoding targeted attack

attack success rate.

In Table 4, we can see that after video compression and encoding, the attack success rate of sparse attacks has been greatly reduced, and our method has a very good effect on resisting encoding and the attack performance after encoding has almost no decline.

Comparing Table 2 and Table 4, we can find that the performance of the target attack drops more significantly after video compression encoding. We think this is because the targeted attack is more difficult than the untargeted attack. In order to make the adversarial sample finally classified into the label we selected, the required perturbation will be more refined. It is worth mentioning that in the untar-

geted attack, we selected the label with the second-highest classification confidence in the original output as the optimization target, which also reduced the difficulty of the untargeted attack to a certain extent. When the required disturbance is finer, the video compression coding will do more damage to the added perturbation. This is very easy to understand. Any slight change may affect the final attack effect. In the untargeted attack, because we choose the label with the second-highest confidence level as the optimization target, the attack becomes easier at the same time. The ability to resist video compression encoding will also become stronger. This is because the original model's judgment for this sample is very close to the selected label, and the impact of additional video compression coding can be largely ignored.

### 4.2.3  Visual concealment:

In this section, we will show and compare the visual effects of the adversarial examples we generated.

In Figure 2, from left to right is the clean image, FGSM, PGD, sparse attack, and our method. The first line is the adversarial examples and clean sample, and the second line is the added perturbation, the third line is the adversarial examples after video encoding. For the visual effect, we will convert the perturbation in the floating-point number space to RGB space, and all perturbations less than 0 are displayed with their absolute value. And the final line is the difference between a non-coded image and a coded image, and we also converted it to RGB space for better visual. Since the added perturbation is too small, we must expand each perturbation by five times to improve its visual effect, and we gray it out to get a more obvious contrast effect. We also expand the difference image between the coded frame

and non-coded frame by five times.

It can be clearly seen from the figure that the adversarial examples generated by our method are superior to the previous method in terms of the sparsity of the space and the magnitude of the adversarial perturbation. It is worth noting that before and after the video encoding, the image data has undergone significant changes and differences, and this difference is the cause of the attack ability loss of the adversarial sample. But in our adversarial examples, this phenomenon did not happen. This is because our adversarial examples optimized this encoding loss during the generation process, minimizing the difference before and after encoding and improving the robustness of the adversarial examples.

#### 4.2.4   Multimedia transmission experiment

In this section, we will show the life cycle problem of adversarial video examples in the case of network multimedia transmission. When a video is spread on the Internet, for example, uploaded to video websites such as YouTube and Tiktok, or shared through social software such as WeChat and Facebook, these websites and software will compress and encode the video, which can reduce bandwidth requirements.

We tested the impact of video transmission via WeChat, YouTube, and TikTok on adversarial video examples. For WeChat, we use one user to send video adversarial examples to another user and download the adversarial examples for testing. For YouTube and TikTok, We first upload adversarial examples, then cache them through the download function on the web, and then test these videos. In our tests, we paid particular attention to the performance variation of targeted attacks, as this attack is most affected by the transmission of video compression encoding. The results of the experiments are shown in the table5.

We selected 100 videos that have been successfully attacked and performed video action recognition on them again after being transmitted through WeChat. It can be seen from the table that when the adversarial examples without robustness enhancement face multimedia network transmission, their attack ability will be greatly reduced.

## 5   CONCLUSIONS

In this paper, we explored the impact of video compression encoding on video adversarial examples and proposed a method for generating video adversarial examples that can resist attack ability against video compression encoding. Our algorithm is a kind of optimization-based method, and we take the magnitude of added perturbation, class loss, and loss caused by video compression encoding as the optimization goal. A series of experiments on the UCF101 data

| Attack Methods | Target Model | Transmission Methods | Success Rate after Transmission/% |
|---|---|---|---|
| Sparse Attack | R3D | WeChat | 5 |
| | | YouTube | 17 |
| | | TikTok | 8 |
| | R(2+1)D | WeChat | 23 |
| | | YouTube | 29 |
| | | TikTok | 21 |
| Ours | R3D | WeChat | **83** |
| | | YouTube | **77** |
| | | TikTok | **69** |
| | R(2+1)D | WeChat | **92** |
| | | YouTube | **79** |
| | | TikTok | **85** |

Table 5: Multimedia transmission experiment

set show that video adversarial examples are vulnerable to video compression encoding, and our method is superior to the previous works in terms of visual concealment, attack success rate, and ability to resist video encoding. In the future, we will explore in two aspects. First, using the characteristics of video compression encoding to detect and defend against video adversarial examples. Second, we can integrate the characteristics of various video compression encoding methods to design a universal method for generating video adversarial examples.

## 6   ACKNOWLEDGMENT

This work is funded by the Nature Natural Science Foundation of China(62002220). Thanks for the help of Shield Lab of Huawei Technologies Co. Xinghao Jiang is the corresponding author.

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
