# OpenReview forum: "Research on video adversarial attack with long living cycle"
_auai.org/UAI/2022/Conference — UAI 2022 Poster_

### Official Review · Reviewer_5iru · 2022-03-22

**Q2(1) Originality/Novelty:** 1
**Q2(2) Significance/Impact:** 2
**Q2(3) Correctness/Technical Quality:** 2
**Q2(6) Clarity Of Writing:** 3
**Q6 Overall Score:** 3
**Q8 Confidence In Your Score:** 4

**Q1 Summary And Contributions:**

This work is motivated by that the videos often go through many encoding and decoding steps in its transmission, the effect of adversarial attacks becomes diminishing. Thus, it tries to propose a novel attack method whose performance resists such video compression coding. This work evaluate the effectiveness of its method on the UCF-101 dataset with two video classification CNNs (I3D and R(2+1)D.

**Q10 Ethical Concerns (Optional):**

No critical concern.

**Q2 Assessment Of The Paper:**

More detailed information regarding each of these aspects is given below:

**Q2(4) Quality Of Experiments (Optional):**

1: Poor: The experimental evaluation is flawed or the results fail to adequately support the main claims.

**Q2(5) Reproducibility:**

2: Fair: Key resources (e.g., proofs, code, data) are unavailable but key details (e.g., proof sketches, experimental setup) are sufficiently well-described for an expert to confidently reproduce the main results.

**Q3 Main Strengths:**

+ This work addresses an interesting problem of how to make adversarial attack robust in the process of video compression coding.

+ For this purpose, it proposes a new optimization-based attack method named Anti-encoding Adversarial Attack (AEAA).

+ The proposed AEAA shows better performance over some basic adversarial attack methods on the UCF-101 dataset.


**Q4 Main Weakness:**

1. The technical novelty of this work is unclear.
- The proposed approach is rather incremental as the derivation of new attack method in Eq.(5) is rather straightforward from previous work.
- Fig.1 depicts the proposed method, but it is too rough. It should be re-drawn in a way  to clarify what the novelty is.

2. The experimental results are weak.
- This work evaluates with only a single dataset – UCF-101.
- It also compares with only basic baselines: FGSM, PGD and Sparse attack.
- Thus, more thorough evaluation should be performed with more CNN architectures in more datasets against more recent baselines.

**Q5 Detailed Comments To The Authors:**

3. Survey on previous work is too cursory.
- Given that video classification has been studied much in computer vision community, the survey in section 2.1 is too sketchy.
- Section 2.2 has the same issue; only three basic adversarial attack methods are mentioned (e.g., FGSM, PGD and C&W algorithm), and only two papers are cited for video adversarial attack works (e.g., (Wei et al. 2019) and (Chen et al. 2021)).

**Q7 Justification For Your Score:**

Please refer to #Q3-4.

**Q9 Complying With Reviewing Instructions:**

1: Yes.

---

### Official Review · Reviewer_o5Sz · 2022-03-31

**Q2(1) Originality/Novelty:** 2
**Q2(2) Significance/Impact:** 2
**Q2(3) Correctness/Technical Quality:** 3
**Q2(6) Clarity Of Writing:** 2
**Q6 Overall Score:** 5
**Q8 Confidence In Your Score:** 3

**Q1 Summary And Contributions:**

This paper identifies the attack degradation problem after encoding and decoding videos. While current methods are vulnerable to this degradation, this paper offers an easy-to-implement solution to combat this issue. The basic idea is to re-train the model until attack succeeds. The proposed solution has been evaluated on UCF-101 dataset and shown superiority against attack degradation after encoding and decoding.

**Q2 Assessment Of The Paper:**

More detailed information regarding each of these aspects is given below:

**Q2(4) Quality Of Experiments (Optional):**

3: Good: The experimental evaluation is adequate, and the results convincingly support the main claims.

**Q2(5) Reproducibility:**

3: Good: Key resources (e.g., proofs, code, data) are available and key details (e.g., proofs, experimental setup) are sufficiently well-described for competent researchers to confidently reproduce the main results.

**Q3 Main Strengths:**

- The perspective is novel and interesting. I believe the attack degradation problem after encoding and decoding videos is a rarely-touched yet important topic in the adversarial attack area. This paper offers some new insights into this issue.
- The proposed method is simple yet effective.
- The experiments seem convincing.

**Q4 Main Weakness:**

- Despite being effective, the proposed method is more of a heuristic solution, without in-depth analysis of how this degradation emerges and how we can get rid of them.
- Some important details are missing. See Q5 below.
- The compared methods are somewhat old. Are there any new methods that can be involved in the experiments, e.g., [Chen et al., 2021] mentioned in Related Work?
- Since the proposed solution relies on an intrinsic encoding and decoding method, I was wondering its ability to generalized to different encoding/decoding methods which is likely to encounter in real-life situations.
- The writing of this paper should be further polished.

**Q5 Detailed Comments To The Authors:**

- The authors mentioned in Intro that they "use the gradient information of the sample to update the next round". How exactly this gradient information is utilized, and how does it help? I prefer to see these being carefully discussed in the main body of this paper since they are the essence of the proposed solution.
- Why Eq. (2) is in this specific form? Also, why choosing the label with second highest confidence score, instead of the third, or the second plus the third? What is the intuition/motivation behind these choices? I hope these would be discussed in detail.
- Typo: "This is easy to understand" in page 4.

**Q7 Justification For Your Score:**

This paper offers some new insights into the attack degradation problem after encoding and decoding videos, yet lacks careful discussion of its motivation. Also the technical contribution is somewhat limited.


**Q9 Complying With Reviewing Instructions:**

1: Yes.

---

### Official Review · Reviewer_ZApM · 2022-04-14

**Q2(1) Originality/Novelty:** 3
**Q2(2) Significance/Impact:** 3
**Q2(3) Correctness/Technical Quality:** 3
**Q2(6) Clarity Of Writing:** 4
**Q6 Overall Score:** 8
**Q8 Confidence In Your Score:** 4

**Q1 Summary And Contributions:**

This paper provides a method for creating adversarial attacks on videos that can survive video compression. They show that their method works much better than standard attacks after video compression. Standard approaches, both targeted and untargeted, do not work well after compression, especially in the case of targeted attacks.

I have read the author's responses to my own and the other reviews. I have noted above about why I think the other reviews are overly harsh. I still vote to accept.

**Q2 Assessment Of The Paper:**

More detailed information regarding each of these aspects is given below:

**Q2(4) Quality Of Experiments (Optional):**

4: Excellent: The experimental evaluation is comprehensive and the results are compelling.

**Q2(5) Reproducibility:**

4: Excellent: Key resources (e.g., proofs, code, data) are available and key details (e.g., proof sketches, experimental setup) are comprehensively described for competent researchers to confidently and easily reproduce the main results.

**Q3 Main Strengths:**

The work is well-motivated, and very successful.
The experiments are thorough.
The paper is very well written, with just minor edits needed.

**Q4 Main Weakness:**

One weakness is that it is unclear why the proposed method results in such smaller changes to the videos. This requires more explanation.

**Q5 Detailed Comments To The Authors:**

Thank you for writing such a clear and strong paper! It was easy to read and review.

My main question (as noted above) is why the perturbations created by your method result in so much smaller changes to the video. Is it because you always picked the output that was second highest?

Minor comments:

First, the title is not appealing:
Research on video adversarial attack with long living cycle doesn’t express the result very well, and “long living cycle” doesn’t evoke a schema in the reader’s head. I suggest something much more straightforward, such as:

An adversarial video attack that survives video compression

The abstract is also quite long. Try to pare it down. For example, the third and fourth sentences could be removed, and the beginning of the fifth. Also, the first sentence is a little odd. I suggest rewriting it as:

In recent years, a great deal of research has been focused on the vulnerability of deep networks to adversarial attacks. In particular, very successful methods have been proposed to attack networks that process videos. However, the impact of video compression on adversarial attacks has not received much attention. Here we show that video compression severely reduces the performance of existing attack methods, suggesting that video compression is an effective defense against such methods. We then propose a video adversarial attack method that is robust to compression, in both targeted and untargeted attacks, taking into account rounding loss in normalization, and that operates on the original integer domain. We perform extensive experiments demonstrating that our method achieves high attack success rates both before and after compression and decompression, and uses much smaller perturbations in the untargeted case.

Minor comments:

Deep Neural Networks (DNNs) has achieved ->
Deep Neural Networks (DNNs) have achieved

in the fields of images classification ->
in the fields of image classification

makes people begin to think of ->
has demonstrated

such like image classification ->
such as image classification

You mention gaussian noise and then call it white noise. These are very different!

with very high transfer-ability. ->
with very high transferability.

They only test their videos in floating number domain. That means examples generated by these methods cannot transfer stability. ->
They only test their methods in the normalized, floating point representation of the video, raising the issue of whether these methods can survive standard video encoding schemes.

denotes a clean video sample ->
denote a clean video sample

The y_truth is the truth label of sample X ->
Here, y_truth is the ground truth label of sample X

the optimization function can converse more quickly. ->
the optimization function can converge more quickly.

if we storage our adversarial example by video encoding, ->
if we store our adversarial example using video encoding,

To be simple, the lossless steps in the process of encoding will not be showed in our algorithm. ->
To simplify our presentation, the lossless steps in the process of encoding will not be included in the description of the algorithm.

This is easy to understand, This is easy to understand. ->
This is easy to understand,

This causes a great loss of information, shown in the image is the reduction of color space and the lack of details. ->
This causes a great loss of information in the image, reducing the color space and removing detail.

X ′ is different with X ->
X ′ is different from X

We chose the label ->
We choose the label

It is easy to reconstruct our algorithm. And achieved similar accuracy rates to the reputation of proposing these model papers. ->
Our algorithm is easily reproduced (we will publish our code after acceptance). Our experiments with standard methods achieved similar accuracy rates as those reported in the original papers.

Metrics: refer to the sparse attack paper Wei et al. [2019], we use four metrics to evaluate various aspects. ->
Metrics: we follow the methodology in the sparse attack paper Wei et al. [2019], and use four metrics to evaluate various aspects of model performance. [But then you list five metrics!]

However, this doesn’t main they have high resistance. ->
However, this doesn’t mean they have high resistance.

Although we can use this coded sample to attack success, but the robustness of adver- sarial example has been destroyed. ->
Although we can use this coded sample to successfully attack the video, the result of the attack is different than in the original domain, making it less consistent.

the predicted label of the video sample will deviate to the unknown place. ->
the predicted label of the video sample will change in unknown ways.

While the rounding loss is caused, ->
When there is the rounding error,

Comparing Table 2 and Table 4, we can find that ->
Comparing Table 2 and Table 4, we see that

we also converted it to RGB space for better visual. ->
we also converted it to RGB space for better visualization.

these websites and softwares ->
these websites and their software

the experiments are shown in the table5. ->
the experiments are shown in Table 5.


**Q7 Justification For Your Score:**

The algorithm is clear, the results are good, the paper is well-written.

**Q9 Complying With Reviewing Instructions:**

1: Yes.

---

### Decision · Program_Chairs · 2022-05-15

**Decision:**

Accept (Poster)

**Comment:**

Meta Review: There was a spread of views on this paper. However the most positive reviewer for this paper made a strong case for it, and the authors addressed various concerns well in their response. All things considered, there is a good case for acceptance.